# Anatomy-to-tract mapping infers white matter pathways without diffusion streamline propagation

Yee-Fan Tan [1,2,3,5], Khoi Minh Huynh [1,2,5], Siyuan Liu [4], Raphaël C.-W. Phan[3], Chee-Ming Ting[3] & Pew-Thian Yap [1,2] ✉

Diffusion tractography, a cornerstone of white matter mapping, relies on point-to-point streamline propagation, a process often limited by the signal-to-noise ratio and spatioangular resolution of diffusion MRI (dMRI). Here, we present Anatomy-to-Tract Mapping (ATM), a framework that generates bundle-specific streamlines directly from T1-weighted MRI without requiring orientation field estimation, voxelwise segmentation, or streamline propagation. ATM leverages the high quality and minimal distortion of anatomical MRI and learns from paired T1w and tractogram data to synthesize anatomically plausible, subject-specific streamlines. This anatomy-driven approach addresses complex configurations such as crossing, kissing, and bending fibers, providing robust bundle reconstructions. Using the TractoInferno dataset with 30 white matter bundles, we evaluate ATM against diffusion-based methods, including MRtrix probabilistic tracking with BundleSeg and SCIL atlas warping. ATM demonstrates strong performance across multiple metrics, including bundle similarity, volume coverage, angular correlation, and geometric fidelity.

Diffusion MRI (dMRI) tractography remains the primary computational technique for in vivo mapping of long-range white matter pathways[1,2], relying on local deterministic[3,4] or probabilistic[5,6] streamline propagation, or global[7–9] streamline reconstruction techniques. However, its accuracy is hindered by several key limitations. Partial volume effects, which occur when multiple fiber bundles with different orientations intersect within a single voxel, complicate fiber orientation estimation and lead to errors in complex configurations such as bending, fanning, crossing, and kissing fibers[10,11]. Another challenge is the bottleneck problem, where multiple fiber populations converge in a narrow region, causing orientation ambiguities that result in false-positive streamlines[12,13]. Moreover, dMRI-based tractography is highly sensitive to data quality, which is often limited by signal-to-noise ratio and spatioangular resolution[14]. The acquisition of high-quality dMRI data remains challenging, particularly in clinical settings, where the

quality of neuroimaging data often renders it unsuitable for reliable tractography.

Recent developments in machine learning (ML) and deep learning (DL) have shown considerable promise in advancing tractography. Several DL approaches have been developed to predict fiber tract orientations from diffusion signals. These include methods based on orientation forecasting, which predict the next location of a point on a fiber streamline based on its previous positions[15–18], as well as reinforcement learning models that trace fibers by optimizing a reward function for accurate tract reconstruction[19]. Additionally, DL models have been used to predict fiber orientation distribution functions, which represent the likelihood of fiber directions at each voxel[20–22], and Fisher-von-Mises distributions to improve tractography results[23].

A notable recent innovation is a DL-based approach utilizing a convolutional-recurrent neural network architecture to approximate

[1]Department of Radiology, University of North Carolina at Chapel Hill, Chapel Hill, NC, USA. [2]Biomedical Research Imaging Center, University of North Carolina at Chapel Hill, Chapel Hill, NC, USA. [3]School of Information Technology, Monash University, Subang Jaya, Malaysia. [4]Marine Engineering College, Dalian Maritime University, Dalian, China. [5]These authors contributed equally: Yee-Fan Tan, Khoi Minh Huynh ✉e-mail: ptyap@med.unc.edu

tractography directly from T1-weighted (T1w) MRI, bypassing the need for dMRI data[14,24]. This method attempts to infer brain connectivity solely from anatomical features, offering a promising alternative to traditional dMRI-based techniques. While this model has shown encouraging results, it still relies on point-to-point tracking, which remains vulnerable to the same tractography challenges, such as fiber crossing and bending, leading to artifacts and inaccuracies in reconstructed pathways.

Generative models have recently emerged as a more robust alternative to conventional streamline propagation. For example, generative sampling in bundle tractography using autoencoders (GESTA)[25] uses an autoencoder-based framework to synthesize streamlines in a bundle-specific manner, eliminating the need to propagate based on local orientation fields. GESTA enhances spatial coverage in hard-to-track or sparsely represented bundles by generating anatomically plausible streamlines post hoc. Another study[26] employs a convolutional variational autoencoder (VAE) to embed streamlines into a low-dimensional latent space and uses kernel density estimation (KDE) to synthesize population-specific bundle templates. This technique captures bundle shape variability more effectively than atlas-based methods and supports direct segmentation from tractograms. Complementary to these generative frameworks are bundle-specific tractography strategies, which explicitly target the limitations of whole-brain tracking. For example, TractSeg[27,28] introduces tract orientation maps along with segmentation masks and endpoint regions to enable accurate, efficient, and reproducible bundle reconstructions without the need for whole-brain tractography or manual dissection. Together, these innovations lay the foundation for more precise and data-driven approaches to white matter mapping.

In this work, we introduce a deep learning framework for Anatomy-to-Tract Mapping (ATM), eliminating the reliance on dMRI. Unlike previous ML and DL approaches[14–16,18–21,23], ATM learns to generate complete, subject-specific white matter bundles directly from anatomical MRI rather than performing point-to-point tracking. By leveraging the high-quality, low-distortion characteristics of T1w MRI, ATM mitigates challenges arising from complex fiber configurations, such as crossing fibers and bottlenecks (Fig. 1). Specifically, ATM captures anatomical context from T1w MRI to guide the reconstruction of full white matter bundles, employing a

probabilistic approach where random vectors are drawn from a probability density function generated via KDE, following a strategy inspired by recent work using VAE for streamline generation[26]. These generated bundles are then refined through filtering and trimming techniques (Fig. 2c).

ATM successfully generates subject-specific, anatomically accurate white matter bundles directly from T1-weighted MRI, producing reconstructions that closely resemble ground truth obtained via conventional dMRI tractography. These findings indicate that full white matter pathways can be reconstructed from anatomical MRI alone, without relying on diffusion-based streamline propagation, providing an alternative approach for white matter mapping.

## Results
### Framework overview
For reproducibility, we used the TractoInferno dataset[29], which includes 30 anatomically defined white matter bundles (Table S1) across 284 subjects. We followed the official dataset partition, allocating 198 subjects for training, 58 for validation, and 28 for testing. T1w images and ground truth bundles, generated using an ensemble of deterministic, probabilistic, particle filtering tractography (PFT), and surface-enhanced tractography (SET) algorithms, followed by manual curation, were used to train ATM. T1w scans were rigidly registered to the MNI152 space[30] and intensity-normalized to [0, 1]. To learn the mapping from T1w images to white matter tracts, 3000 streamlines per bundle were randomly selected and resampled to 128 equidistant points along each streamline using Dipy[31]. Streamlines were then transformed to MNI space using Scilpy (https://github.com/scilus/scilpy). White matter surfaces were extracted from T1w scans with FreeSurfer[32].

ATM (Fig. 2) generates $N$ streamlines per bundle by sampling $N$ 64-dimensional latent vectors from a KDE-based distribution, conditioned on anatomical features from T1w images. For evaluation, $N = 3000$, 6000, and 9000 streamlines were sampled per bundle. Implausible streamlines were removed using the T1w white matter mask, and remaining streamlines were trimmed at the white matter surface. ATM can also infer population-level bundles from T1w images (Fig. 2d). T1w scans from the test set were non-linearly registered to the MNI152 space (1 mm isotropic resolution) using ANTs[33], averaged, and

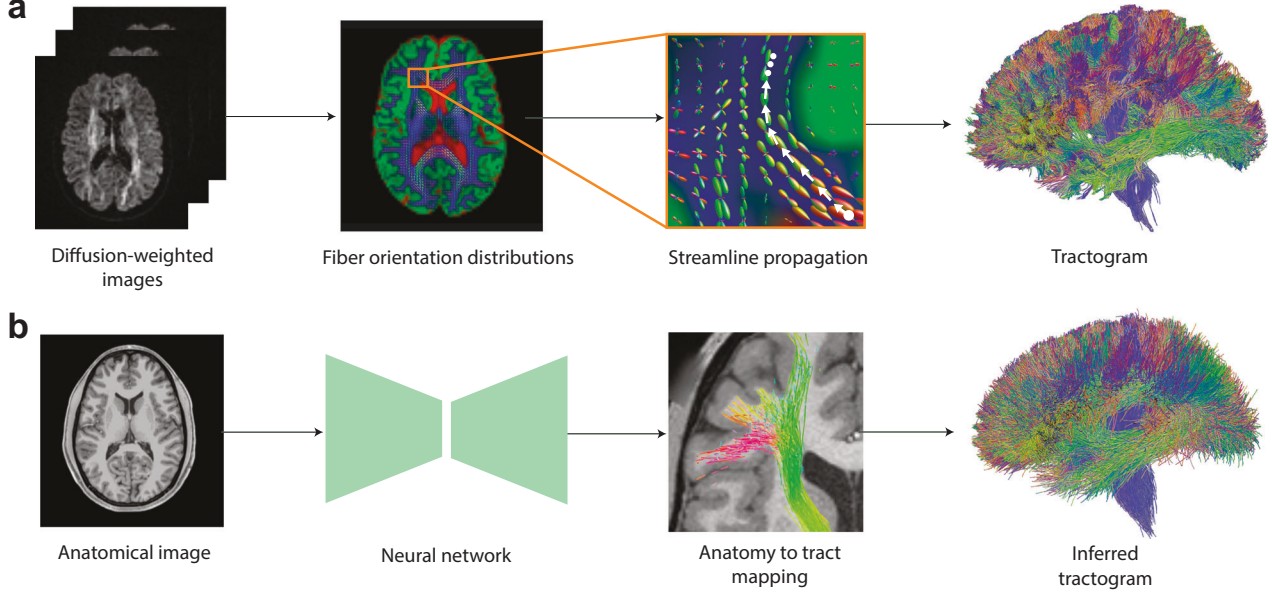

**Fig. 1 | Streamline propagation versus anatomy-to-tract mapping (ATM). a** Conventional tractography propagates streamlines point-to-point from seed points using local fiber orientation distributions, either deterministically or probabilistically. **b** ATM infers entire bundles from an anatomical T1-weighted (T1w) image.

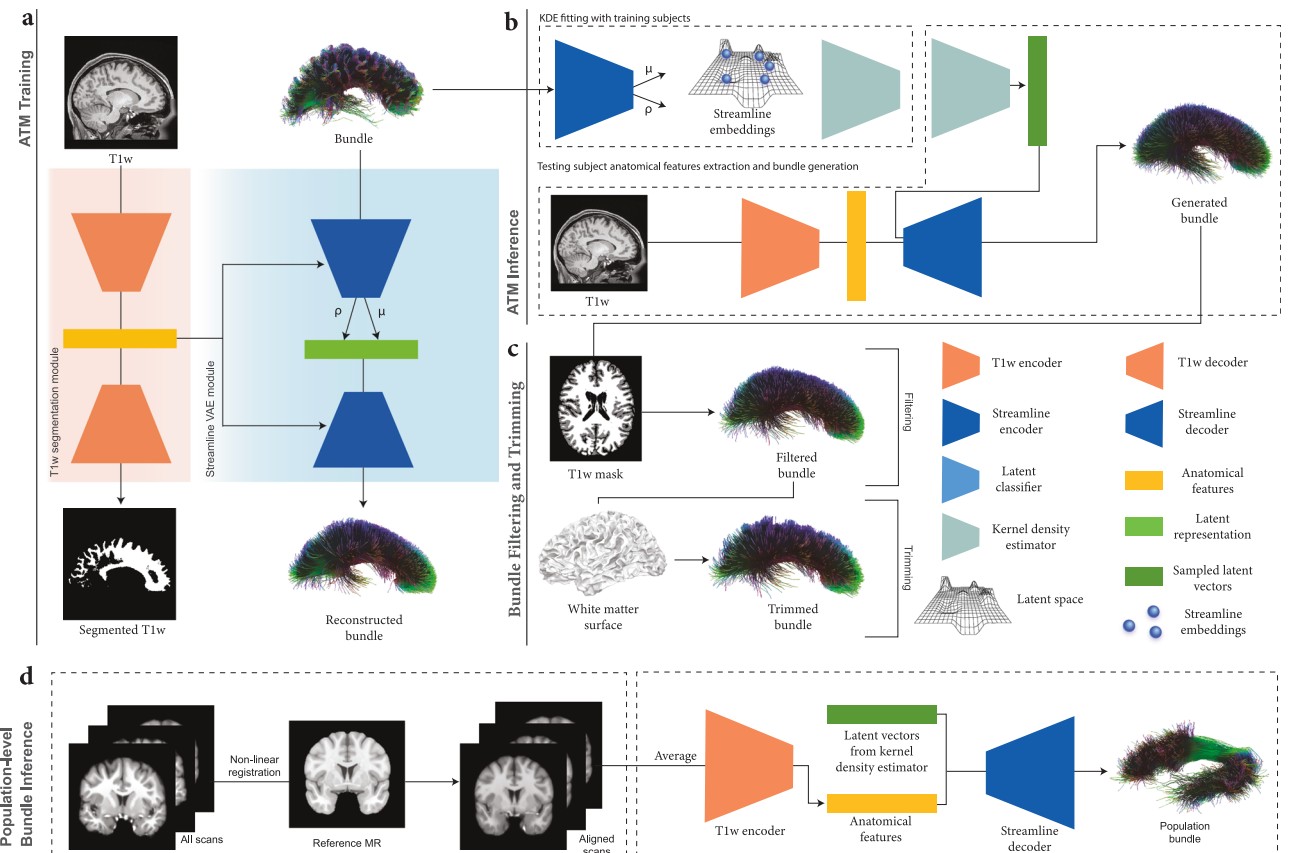

**Fig. 2 | ATM overview. a** ATM model training. ATM takes as inputs anatomical T1w images and streamlines, projects them into different latent spaces through the T1w encoder $E_A$ and bundle encoder $E_S$, respectively. T1w segmentation decoder $D_A$ guides anatomical feature extraction. Bundle decoder $D_S$ reconstructs the output streamlines from the latent space $z_s$, conditioned on the anatomical features $a$ extracted using $E_A$. **b** ATM inference. The probability density function of the streamline embeddings is estimated using KDE. ATM utilizes points sampled from the estimated probability density function, conditioned on anatomical features $a$, to generate streamlines of a particular bundle. **c** Bundle filtering and trimming. The generated bundle is refined by filtering out excess streamlines with guidance from a T1w-derived tissue mask. The filtered bundle is subsequently trimmed based on the white matter surface. **d** Population-level bundle inference. T1w scans from the testing population are non-linearly registered to the MNI152 space at 1 mm isotropic resolution. The registered scans are averaged to create a population template, which the trained ATM model uses to extract anatomical features and infer bundles representative of the population.

processed with ATM's encoder to extract anatomical features for bundle generation.

We evaluated ATM against three representative baselines: (i) MRtrix with BundleSeg, applying BundleSeg to whole-brain probabilistic streamlines generated by MRtrix; (ii) Atlas warping, which transforms the SCIL WM Atlas to individual subject spaces; and (iii) ATM-Population, applying ATM to the population-averaged T1w template. For completeness, results from TractSeg, which defines the bundles differently and is therefore not directly comparable, are provided in Supplementary Materials.

### ATM generates complete bundles with characteristic shapes

We demonstrate the effectiveness of ATM in generating white matter bundles (Table S1) directly from anatomical T1w MRI, without the need for point-to-point tracking as required in conventional tractography. Figure 3 presents tractograms generated for one representative subject, indicating that ATM-generated bundles closely resemble the ground truth. ATM successfully reconstructs complete bundles that other methods might reconstruct inadequately in terms of spatial extent (e.g., CC_Oc and UF_L from MRtrix with BundleSeg in Fig. 3). These results highlight ATM's ability to generate anatomically plausible streamlines based on randomly sampled latent vectors and anatomical features derived from T1w scans. Figures S2–S7 display all bundles generated with each method. Results for TractSeg are shown in Fig. S9a.

ATM can also generate anatomically reasonable bundles from clinical low-field and low-resolution T1w images[34], which were upsampled from 1.6 mm × 1.6 mm × 5 mm and 1.5 mm × 1.5 mm × 5 mm resolutions, respectively, to 1 mm isotropic resolution for bundle inference (Fig. S8).

In addition, the bundles generated with ATM show strong similarity in tract orientation distribution (TOD) to the ground truth, as evidenced by high angular correlation coefficients (ACCs) between TOD maps derived from the ground truth and generated streamlines. As shown in Fig. 4, ATM achieves higher accuracy (ACC) than both population bundles and bundles generated using MRtrix with BundleSeg. For completeness, results from TractSeg are provided in Fig. S9b. This accuracy underscores the anatomical fidelity and overall precision of bundles generated with ATM. Figure S20 presents violin plots of the paired differences in ACC between ATM and the other methods.

### High geometric agreement with ground truth

Using the bundle adjacency (BA) metric from bundle analytics (BUAN)[35], we assessed the generated bundles in terms of shape similarity with respect to the ground truth. Across bundles, mean BA scores across subjects ranged from 0.5883 to 0.9420 (Table S2). We also compared the generated and ground truth bundles in terms of spatial coverage using Tractometer[36,37], focusing on volumetric Dice, overlap, and overreach scores (Table S2). ATM consistently produces bundles

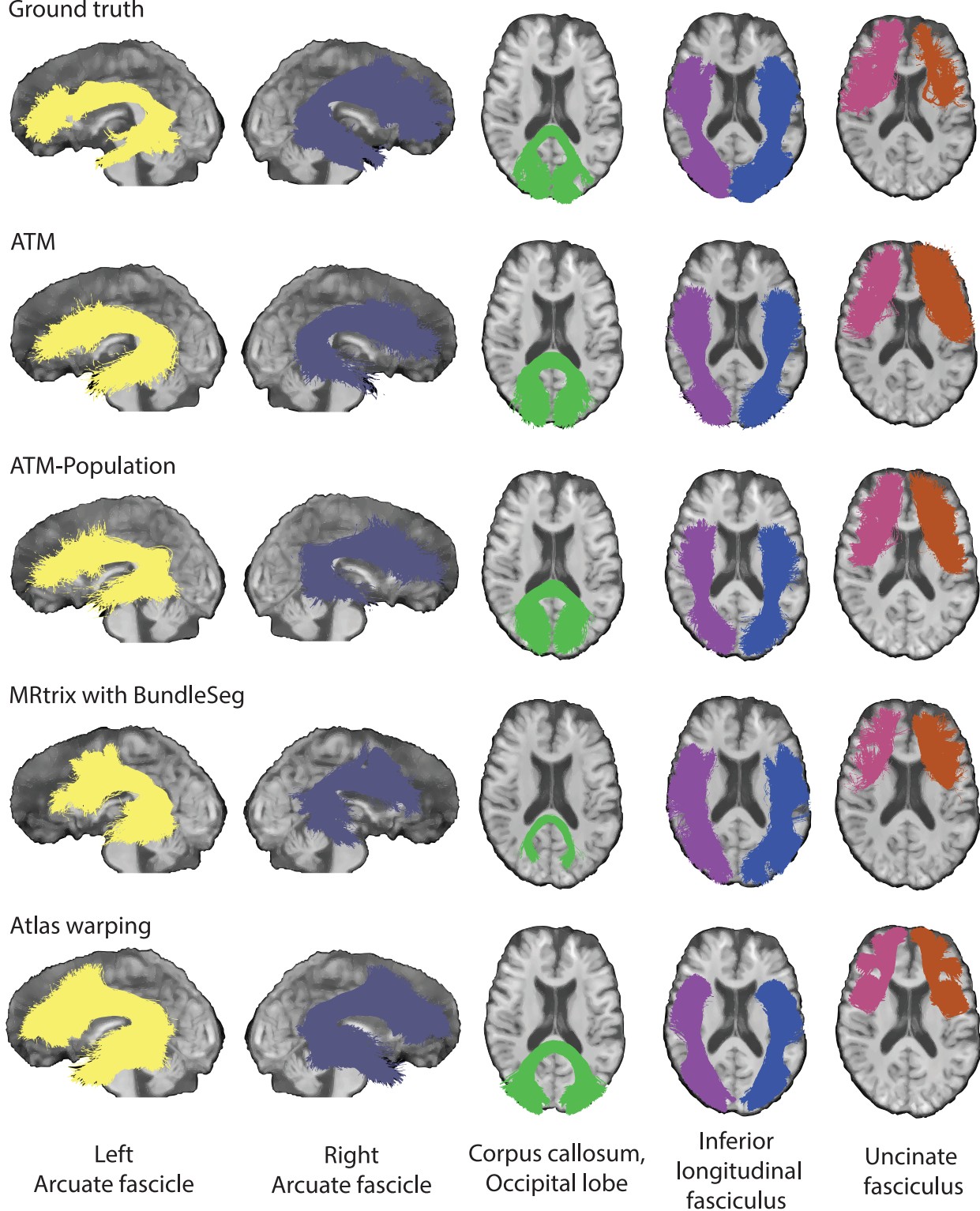

**Fig. 3 | Tractography comparison.** Qualitative comparison of representative bundles reconstructed by different methods. (TractoInferno Subject ID: 1135).

with competitive Dice, overlap, and overreach scores, ranging from 0.3215 to 0.5848, 0.4369 to 0.6785, and 0.3379 to 2.1464, respectively.

ATM yields competitive performance relative to existing tractography methods. The heatmaps in Fig. 5 show paired differences at the individual level between ATM and the competing methods. In particular, ATM outperforms competing methods in terms of BA and Dice scores across most tracts. ATM also shows consistently high overlap scores, suggesting better bundle spatial coverage. These

improvements highlight ATM's ability to capture the diverse shapes and volumes of different bundles, including those with complex trajectories. Notably, ATM yields higher overreach scores for some bundles (e.g., FPT and POPT), particularly when compared to MRtrix with BundleSeg and SCIL WM atlas warping. This increase in overreach is due to the generative nature of ATM. However, the increased overreach is accompanied by higher Dice and overlap scores, indicating that the broader coverage provided by ATM remains

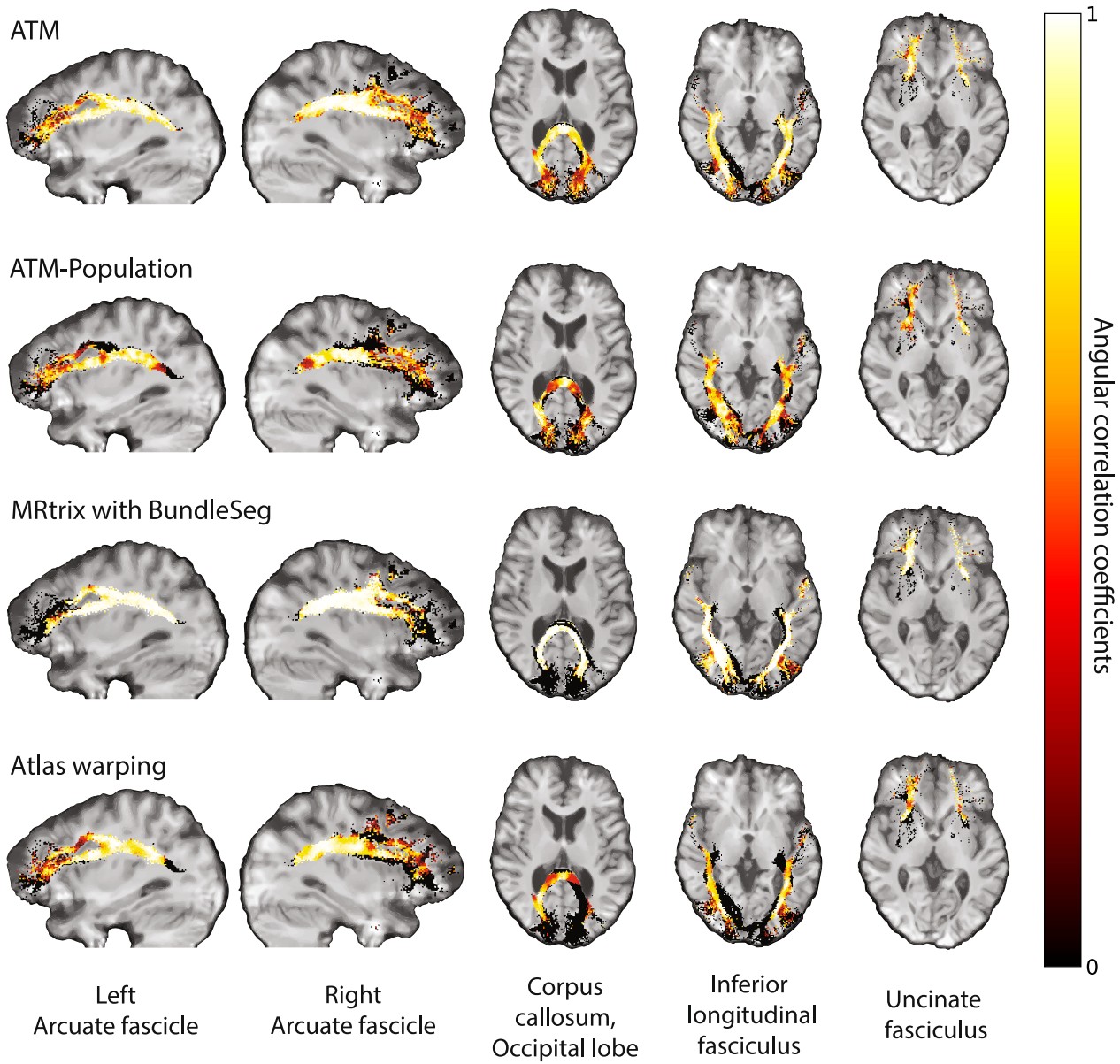

ATM

ATM-Population

MRtrix with BundleSeg

Atlas warping

Left
Arcuate fascicle

Right
Arcuate fascicle

Corpus
callosum,
Occipital lobe

Inferior
longitudinal
fasciculus

Uncinate
fasciculus

Angular correlation coefficients

**Fig. 4 | Orientation alignment.** Angular correlation coefficients (ACCs) were computed between the tract orientation density maps of the ground truth and generated streamlines. (TractoInferno Subject ID: 1135).

anatomically relevant. Interestingly, the ATM-generated population bundles exhibit weaker performance across these metrics, particularly in Dice and BA scores. This drop in accuracy likely stems from its reliance on population-averaged features, which reduces its ability to account for subject-specific anatomical variability, leading to less precise bundle generation. Figures S14–S17 show the violin plots of the paired differences between ATM and the other methods in terms of BA, Dice, overlap, and overreach scores.

ATM consistently performs well on streamline validity metrics assessed using Tractometer[36,37], which evaluate the validity of individual streamlines within each bundle. Results indicate that the generated bundles achieve valid streamline (VS) ratios from 0.5312 to 0.9615 and invalid streamline (IS) ratios from 0.0385 to 0.4678 (Table S2). These results indicate that a substantial majority of ATM-generated streamlines adhere to expected anatomical pathways. As shown in Fig. 5, ATM consistently outperforms baseline methods on both VS and IS ratios, with notably fewer invalid streamlines across most bundles. This indicates that ATM captures anatomically relevant trajectories and avoids common sources of tractography error, such as premature

terminations. Figures S18 and S19 present violin plots of the paired differences in VS and IS ratios between ATM and the other methods.

Bundle geometry was further assessed by measuring mean length, span, volume, and surface area (Table S7). The absolute percent differences (APDs) of these geometric measures with respect to the ground truth range from 6% to 24%, 5% to 23%, 21% to 63%, and 11% to 60%, showing strong morphological agreement between the generated and ground truth bundles. As shown in Fig. 6, ATM consistently yields lower geometric errors relative to the ground truth than competing methods, including MRtrix with BundleSeg, and SCIL WM atlas warping. These improvements are especially notable in volume and surface area APDs, where competing methods often show much larger APDs, particularly in bundles such as the MCP, POPT, and PYT. ATM results with lower APDs suggest a more accurate reconstruction of the overall bundle geometry. Similar to previous findings, the ATM-generated population bundles again show lower accuracy, with higher APDs across all geometric measures. Across these evaluations, ATM consistently outperforms ATM-Population, indicating that subject-specific anatomical cues

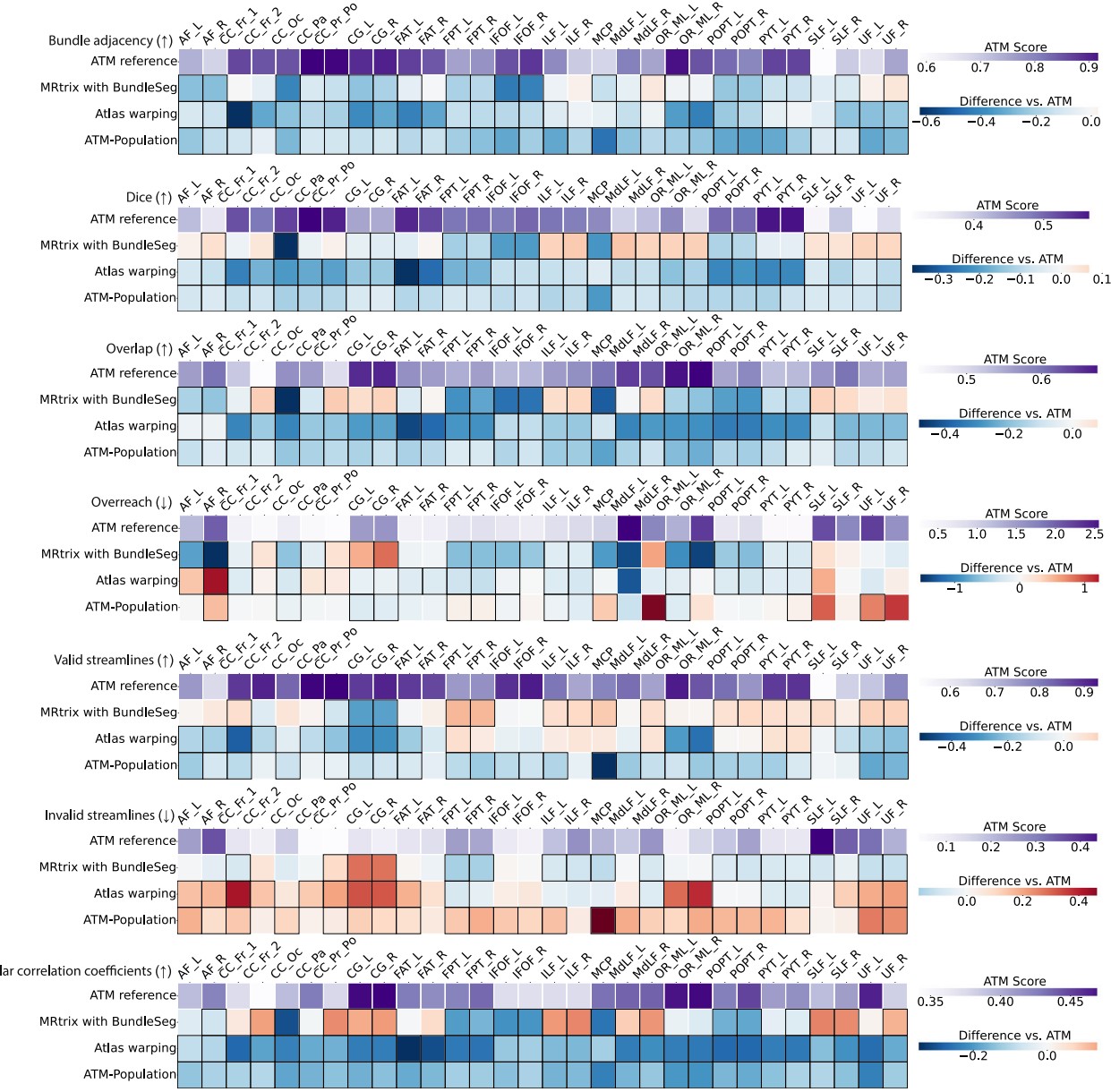

**Fig. 5 | Similarity and coverage of bundles generated with ATM, ATM-Population, MRtrix with BundleSeg, and SCIL WM atlas warping.** The first row of each heatmap shows the ATM values as the baseline. The subsequent rows display the paired performance differences of other methods relative to this baseline. For each metric, the arrow indicates preference: ↑ higher is better and ↓ lower is better. Cells outlined with squares indicate statistically significant differences. Statistical significance was evaluated using a two-sided paired $t$-test ($p < 0.05$). Source data are provided as a Source Data file.

substantially improve the reconstruction of white matter pathways. Violin plots in Figs. S21–S24 further illustrate the paired differences in APD values. Visual and numerical comparisons in Figs. 3, 5, 6, S9, S10, S11, S14–S29 support our observation that ATM generates bundles that are faithful to subject-specific anatomy.

### Improved bundle-specific fidelity in connectomics
We evaluated the connectomes derived from the bundles based on the Desikan–Killiany–Tourville (DKT) atlas[38,39], with edges defined by streamline counts between pairs of the 83 cortical regions. For fair comparison across methods, each connectome was normalized by its total streamline count. To compare against the ground truth connectomes, we used Pearson correlation and graph-theoretic measures, including network density, characteristic path length (CPL), global efficiency, and modularity. Results are summarized in Table S7 and visualized in Fig. 6.

ATM-derived connectomes showed Pearson correlations ranging from 0.28 to 0.61, indicating moderate agreement. Despite relatively high APDs—22–113% for density, 59–132% for CPL, 90–162% for efficiency, and 62–179% for modularity, ATM consistently outperforms MRtrix with BundleSeg and SCIL WM atlas warping across most metrics. The heatmaps in Fig. 6 reveal larger deviations in the connectomes produced by competing methods. ATM also outperforms ATM-Population, highlighting its ability to capture each subject's unique connectomic fingerprint from T1w scans of anatomy. Violin plots in Figs. S25–S29 show paired performance differences between ATM and competing methods.

### Performance in relation to streamline sample size
To assess the impact of streamline density, we compared ATM-generated bundles using 3000, 6000, and 9000 streamlines per bundle. As shown in Fig. S12, a higher streamline count improves BA

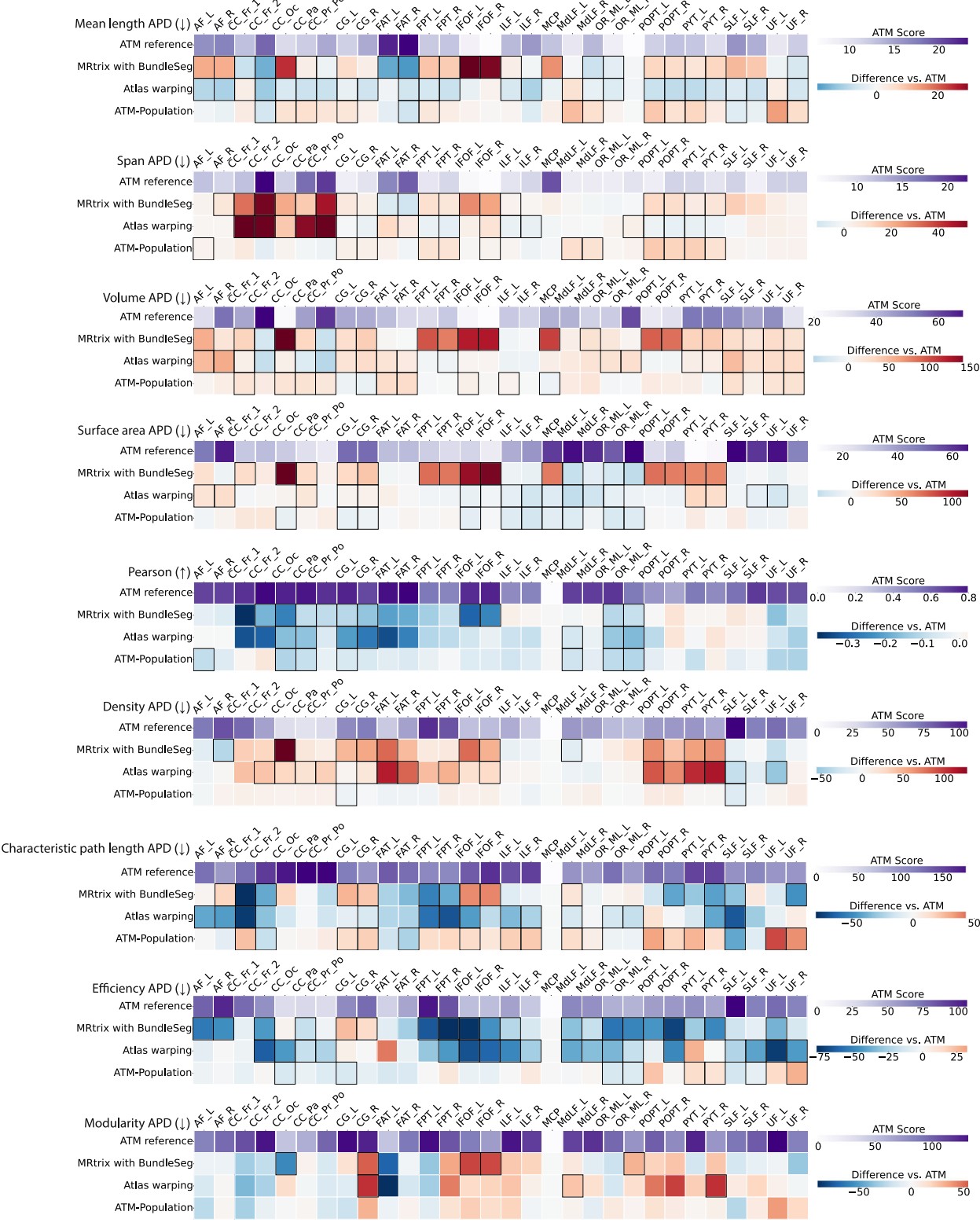

*APD: Absolute percentage difference

**Fig. 6 | Geometry and connectomics of bundles generated with ATM, ATM-Population, MRtrix with BundleSeg, and SCIL WM atlas warping.** The first row of each heatmap shows the ATM values as the baseline. The subsequent rows display the paired performance differences of other methods relative to this baseline. For each metric, the arrow indicates preference: ↑ higher is better and ↓ lower is better. Cells outlined in squares indicate statistically significant differences. Statistical significance was evaluated using a two-sided paired $t$-test ($p < 0.05$). Source data are provided as a Source Data file.

and ACC scores, reflecting better shape similarity and tract orientation alignment. VS ratios remain high and IS ratios decrease slightly, indicating that most added streamlines remain plausible.

The Dice score does not consistently improve: it declines at higher densities as streamlines extend beyond the core anatomy, increasing both overlap and overreach, particularly between 3000 and 6000 streamlines, and increases modestly at 9000 (Fig. S13). This plateau may reflect the saturation of KDE sampling around high-probability regions.

Geometrically, increasing the number of streamlines reduces the APDs for streamline mean length and span but increases the APDs for volume and surface area, consistent with the inflation of bundle boundaries. Connectome Pearson correlations improve with higher streamline counts. However, graph-theoretic measures, such as density, CPL, and global efficiency, improve up to 6000 streamlines but slightly worsen at 9000, likely due to increased network noise from over-connectivity. Overall, while more streamlines tend to enhance shape fidelity and connectivity alignment, excessively high counts may lead to boundary overreach and reduced geometric and connectomic specificity.

## Discussion

We introduced ATM, a deep learning model that extracts anatomical features from T1w images to infer subject-specific white matter bundles. ATM overcomes challenges commonly faced by conventional tractography methods, such as navigating complex configurations like crossing, kissing, and bending fibers, as well as constricting bottlenecks. By generating streamlines directly from anatomical images, ATM reduces the need for high-quality dMRI data and offers an anatomy-driven alternative to diffusion-based tractography. Our results demonstrate a strong resemblance between ATM-predicted bundles and the ground truth. ATM shows competitive performance across multiple criteria, including similarity, spatial coverage, streamline validity, geometric consistency, and connectivity. Compared to several baseline methods, such as MRtrix with BundleSeg and population-based bundle warping, ATM shows improved performance both qualitatively and quantitatively. While methods like MRtrix with BundleSeg may fail to reconstruct certain streamlines or produce anatomically incomplete bundles, ATM effectively avoids these shortcomings by generating complete subject-specific and anatomically plausible streamlines.

Recent advances in DL-based tractography have primarily focused on predicting local fiber orientations (e.g., Learn-to-Track[16]) or modeling fiber orientation distributions (e.g., DeepTract[20] and CTtrack[21]). However, these methods often struggle with tracking errors, particularly in regions with complex fiber configurations. In contrast, GESTA[25] uses an autoencoder-based framework to generate complete streamlines from the latent space of a trained model. While GESTA improves spatial coverage of white matter bundles by avoiding local streamline propagation, its sampling process depends on the quality of seed streamlines and does not account for individual anatomical differences. In contrast, the ATM incorporates T1w images to guide subject-specific streamline generation, allowing a direct mapping between anatomy and tracts. Another recent method, the Convolutional-Recurrent Neural Network (CoRNN)[14], adopts a teacher-student framework. The student model samples streamlines from T1w MRI by mimicking the teacher model, which is trained to propagate streamlines from dMRI. However, CoRNN's point-to-point tracking is still susceptible to the same errors as traditional dMRI-based tractography, especially in regions with complex fiber configurations. In contrast, ATM achieves competitive coverage and accuracy without relying on diffusion-based streamline propagation.

Including the ATM-Population experiment provides important context, allowing direct comparison between subject-specific and population-level predictions. ATM consistently outperforms ATM-Population on both geometric and connectomic metrics, showing that it effectively utilizes subject-specific anatomical information that is lost in population averages. The results highlight ATM's potential as a precise, individualized alternative to population-based tract inference.

Despite its strengths, ATM has limitations that warrant further improvement. While it can generate streamlines that align well with anatomical pathways, it remains susceptible to errors due to its incomplete encoding of the anatomical details required for precise streamline generation. For example, it may miss subtle variations in white matter structure or overlook complex anatomical relationships, resulting in inaccurate outcomes. Although the model performs well in producing subject-specific bundles, further research is needed to better understand what anatomical features ATM captures. This insight could guide future improvements in streamline fidelity.

Another major challenge is the presence of spurious and noisy streamlines in the generated bundles. This issue often arises from the inherent randomness associated with the latent space sampling process. These spurious streamlines may lead to misleading results in subsequent analyses. To minimize the occurrence of these spurious bundles, future iterations of ATM should consider optimizing model architectures and refining loss functions. With careful adjustments, the model can be enhanced to generate cleaner and more reliable streamlines. Moreover, the current filtering algorithm used in ATM could benefit from enhancements through the incorporation of advanced deep learning models, such as Filtering in Tractography using Autoencoders (FINTA)[40]. FINTA uses an autoencoder framework to efficiently filter streamlines by measuring distances between reference streamlines and those under evaluation. Given that ATM also relies on an autoencoder architecture, integrating FINTA's latent space filtering approach could be straightforward. This integration would enhance ATM's filtering capabilities, leading to higher-quality bundle reconstructions.

ATM is currently trained on bundles and T1w MRI data obtained from healthy individuals. This could reduce the model's effectiveness when applied to individuals with distinct neuroanatomical features or pathologies. For instance, individuals with neurological disorders or developmental abnormalities often exhibit atypical white matter configurations that are not well-represented in the training data. As a result, ATM may struggle to generate accurate white matter bundles in these populations, potentially compromising its clinical utility and reliability in understanding white matter pathways in the presence of pathologies. Fortunately, an ATM could easily incorporate a conditioning mechanism that allows it to be retrained using data reflecting anatomical variations associated with abnormal conditions. By incorporating data from individuals with different pathologies, ATM can learn how to streamline trajectories that adapt to these unique anatomical characteristics. This adaptability is crucial for improving the model's performance across diverse populations, making it more versatile and clinically relevant.

ATM can be enhanced by incorporating additional imaging modalities, such as T2-weighted (T2w) MRI. Combining T1- and T2-weighted data offers a more complete view of brain anatomy, enabling ATM to leverage complementary information. For instance, T2w images can provide insights into fluid-filled structures and changes in tissue properties that T1w images alone may not capture. This approach could improve the accuracy of generated white matter bundles and refine the model to reduce tracking errors, thereby enhancing its overall reliability. Moreover, the ATM architecture can be readily adapted to incorporate diffusion-derived features in future work. This opens the possibility of using the ATM as a general framework that integrates both structural and diffusion MRI, providing an alternative to conventional step-wise streamline propagation and addressing the known limitations of local tractography algorithms.

Inferring white matter pathways from anatomical T1w MRI marks a notable shift in brain connectivity mapping. Traditionally, tractography has relied on dMRI to capture microstructural features such as fiber orientation and anisotropy, typically at scales of micrometers. In contrast, T1w images encode macrostructural anatomy at the millimeter scale, including cortical folding, tissue boundaries, and brain morphology. Our findings suggest that deep learning models can learn meaningful associations between these large-scale anatomical features and the underlying white matter pathways, enabling plausible streamline generation even without explicit diffusion information. This raises important questions: To what extent is the information required for tractography implicitly embedded in anatomical structure? And does tractography, even when diffusion-based, partly reflect macrostructural patterns more than previously acknowledged? Recent studies have hinted at this possibility, and our results further support the notion that anatomical context contributes meaningfully to inferring connectivity. While this does not replace the rich microstructural insights offered by diffusion imaging, it reframes tractography as a multimodal phenomenon: one that bridges spatial scales and imaging modalities. Further research is needed to clarify the individual contributions of different factors and identify the best way to integrate information from multiple sources.

## Methods

The data used in this study were approved by the Institutional Review Boards of all participating sites and complied with relevant ethical regulations. Written informed consent was waived due to the retrospective use of fully de-identified MRI data. Sex and gender information were not available in the de-identified datasets and were not considered in the study design. As this work focuses on methodological development rather than biological or clinical group differences, no sex-stratified analyses were performed. No participant compensation was involved, as no new data were collected for this study.

### ATM architecture

The ATM network (Fig. 2) consists of two main components:

1. A segmentation module (Fig. 2a) with a volumetric image encoder $E_A$ to extract anatomical features $a$ from a T1w image $x$ and a decoder $D_A$ to predict the binary segmentation map $\tilde{x}$ of the bundle of interest.
2. A streamline variational autoencoder (VAE) module (Fig. 2a) that employs an encoder $E_S$ to embed streamline information into a low-dimensional space conditioned on the anatomical information $a$ using FiLM layers[41]. The latent sample, $\mathbf{z}_s$, and anatomical features, $a$, modulated by FiLM layers, are fed into the decoder $D_S$ to reconstruct the target streamline $\tilde{s}$.

See Fig. S1 for the architectural details of ATM.

As depicted in Fig. 2a, for a given input T1w image $x \in \mathcal{X}$ ($\mathcal{X}$ denotes the space of T1w anatomical images) and a streamline $s \in \mathcal{S}$ ($s = \{(x_i, y_i, z_i)\}_{i=1}^K$ represents a sequence of 3D spatial coordinates along the streamline and $\mathcal{S}$ is the space of all such streamlines), the image encoder $E_A$ extracts anatomical features $a = E_A(x)$. Using these high-level anatomical features, the anatomical image decoder $D_A$ reconstructs the corresponding binary segmentation map $\tilde{x} = D_A(a)$. Conditioned on $a$, the streamline encoder $E_S$ outputs the mean vector $\boldsymbol{\mu}_s$ and standard deviation vector $\boldsymbol{\sigma}_s$. A latent vector $\mathbf{z}_s$ is then sampled via reparameterization $\mathbf{z}_s = \boldsymbol{\mu}_s + \boldsymbol{\epsilon} \odot \boldsymbol{\sigma}_s$, where $\boldsymbol{\epsilon} \sim \mathcal{N}(0, \mathbf{I})$ and $\odot$ denotes element-wise multiplication. Finally, the streamline decoder $D_S$ reconstructs the streamline as $\tilde{s} = D_S(\mathbf{z}_s, a)$.

The segmentation module of ATM undergoes training with the primary objective of minimizing the segmentation loss, $\mathcal{L}_{SEG}$, which measures the dissimilarity between the ground truth segmentation $\hat{x}$

and the predicted binary segmentation $\tilde{x}$ of input image $x$. This optimization is achieved through the utilization of the Dice loss function:

$$\mathcal{L}_{SEG} = 1 - 2\left(\frac{\sum_m^M (\hat{x}_m \cdot \tilde{x}_m) + e}{\sum_m^M \hat{x}_m^2 + \sum_m^M \tilde{x}_m^2 + e}\right), \tag{1}$$

where $e$ denotes the smoothing effect and $m$ indicates the $m$th voxel of the image. In the context of streamline reconstruction and generation, ATM incorporates the VAE loss function[42,43]:

$$\mathcal{L}_{VAE} = \frac{1}{N}\sum_{n=1}^N \left[\| s_n - \tilde{s}_n \|^2 + \beta D_{KL}(q(\mathbf{z}_n|s_n) \| \mathcal{N}(0, \mathbf{I}))\right], \tag{2}$$

where $\beta$ is a regularization coefficient that constrains the latent bottleneck, $p(\mathbf{z})$ is the Gaussian prior distribution $\mathcal{N}(0, 1)$, and $D_{KL}(\cdot \| \cdot)$ is the Kullback-Leibler divergence. The first term in the equation represents the mean squared error loss, evaluating how effectively ATM can reconstruct the input streamlines. The second term regularizes the latent space $q(\mathbf{z}_s|s)$ to follow the prior distribution $p(\mathbf{z})$. To ensure the quality of the generated streamlines, ATM explicitly integrates a regularization term $\mathcal{L}_{ADJ}$ that keeps adjacent points on a streamline closely spaced, to prevent spatial dispersion and to ensure geometric smoothness. This is achieved by calculating the mean and maximum of the average Euclidean distances between adjacent points across all streamlines:

$$\mathcal{L}_{ADJ} = \frac{1}{N}\sum_{n=1}^N B(s_n) + \max_n B(s_n), \tag{3}$$

where $B(s_n) = \frac{1}{K-1}\sum_{j=1}^K \| \mathbf{p}_j^{(n)} - \mathbf{p}_{j-1}^{(n)} \|$ is the average distance between consecutive points along the $n$-th streamline, and $\mathbf{p}_j^{(n)}$ denotes the $j$-th point on the streamline. The first and second terms are the mean and maximum of these average distances across all $N$ streamlines. $N$ is the number of streamlines in the batch, and $K$ is the number of points per streamline.

The ATM model is optimized with the following total loss function, weighted by tuning parameters $\lambda_{SEG}$, $\lambda_{VAE}$, and $\lambda_{ADJ}$:

$$\mathcal{L}_{ATM} = \lambda_{SEG}\mathcal{L}_{SEG} + \lambda_{VAE}\mathcal{L}_{VAE} + \lambda_{ADJ}\mathcal{L}_{ADJ}. \tag{4}$$

### Streamline inference

The probability density function (PDF) of the streamline embedding $\mathbf{z}_s$, extracted via the trained streamline encoder $E_S$, is determined using kernel density estimation (KDE) based on the training data. For bundle-specific generation, the streamline decoder $D_S$ reconstructs a streamline $\tilde{s}$ by mapping a latent vector $\mathbf{z}_r$ that is sampled from the estimated PDF, conditioned on anatomical features $a$ extracted from the input T1w image $x$ using the encoder $E_A$.

### Streamline filtering and trimming

We apply the following steps to remove spurious or anatomically implausible streamlines: (i) Discard streamlines that extend outside the brain, based on the tissue segmentation map derived from the T1w image. (ii) Remove streamlines shorter than 20 mm. (iii) Trim streamlines by removing any points that extend beyond the white matter surface mesh, except for streamlines that connect to the brainstem or cerebellum, or those that traverse hemispheres.

### Implementation details

ATM was implemented using PyTorch[44]. The segmentation module was trained using the Dice loss function[45] applied to binary voxel-wise tract occupancy maps. The streamline variational autoencoder (VAE)

was trained with the mean squared error (MSE) loss on streamline coordinates, following the $\beta$-VAE framework[42], which includes a tunable hyperparameter $\beta$ to balance reconstruction accuracy and the learning of statistically independent latent factors. Training was carried out using the Adam[46] optimizer with a learning rate of $1 \times 10^{-4}$ for a total of 1000 epochs. The models with the lowest validation loss over the training epochs were used for inference and evaluation. The weights of the loss functions were $\lambda_{SEG} = 1$, $\lambda_{VAE} = 1$, $\lambda_{ADJ} = 1$, and $\beta = 1$.

### TractoInferno

TractoInferno[29] is a large-scale, open-source tractography dataset specifically curated for training and evaluating machine learning algorithms for white matter tract reconstruction. It comprises 284 subjects (18–75 years of age) scanned on 3T MRI systems across six imaging centers, incorporating both research-grade and clinical-like acquisitions. The dataset includes T1w images, single-shell dMRI, spherical harmonics, fiber orientation distributions (FODs), and high-quality reference streamlines for 30 white matter bundles generated using four different tractography algorithms, followed by extensive manual curation and quality control. RecoBundlesX[47,48] was used for bundle extraction from whole-brain tractograms. ATM was evaluated on 28 subjects, while the remaining subjects were used for model training.

### Clinical and low-field data

We further evaluated ATM on T1-weighted scans from an external clinical dataset of low-field MRI[34]. The dataset comprises 87 neurological scans from 65 participants (20 males, 45 females; aged 19–65 years), acquired at 64 mT with TR = 880 ms, TE = 5.03 ms, 1.6 mm in-plane resolution, and 5 mm slice thickness. Additionally, the dataset contains 11 paired scans acquired at 3T with TR = 10 ms, TE = 4.58 ms, 1.5 mm in-plane resolution, and 5 mm slice thickness. All scans were collected at the Leiden University Medical Center (LUMC). The dataset was collected in accordance with ethical regulations approved by the hospital ethics committee of Leiden Den Haag Delft (approval number NL83272.058.22). Volunteers were recruited through the LUMC internal volunteer system, and a signed Volunteer Information Form (VIF) was obtained prior to each session. Before ATM inference, all images were resampled to 1 mm isotropic resolution.

### Evaluation metrics

ATM-generated bundles were compared to TractoInferno ground truth across six types of metrics: similarity, coverage, validity, orientation, geometry, and connectivity:

1. *Similarity*: We assessed bundle shape similarity using the bundle adjacency (BA) metric from BUAN[35], which quantifies the distance between bundles and ranges from 0 (no similarity) to 1 (perfect match).
2. *Coverage*: Following the Tractometer framework[36,37], we evaluated bundle accuracy using Dice, overlap, and overreach scores. The overlap score quantifies the proportion of ground truth voxels traversed by valid streamlines, while the overreach score quantifies the proportion of non-ground truth voxels mistakenly traversed by these streamlines. The Dice score is computed as the harmonic mean of the overlap and overreach scores.
3. *Validity*: We computed valid streamline (VS) and invalid streamline (IS) ratios, as defined by the Tractometer framework. VS is the proportion of streamlines matching ground truth bundles, and IS is the proportion outside all valid bundles.
4. *Orientation*: We assessed local directional agreement using the angular correlation coefficient (ACC) computed from tract orientation distribution (TOD) maps generated with MRtrix[49], represented in spherical harmonics. ACC was calculated voxel-wise between the predicted and ground truth TODs, restricted to voxels where the ground truth TOD was non-zero[50].

5. *Geometry*: Bundle shape was characterized using DSI Studio[4,51]:
   - Mean length: Average streamline length.
   - Span: Average distance between streamline endpoints.
   - Volume: Total bundle volume.
   - Surface area: The total surface area of a bundle, estimated from the boundary voxels of its volumetric representation.
6. *Connectivity*: Bundle-specific structural connectomes were constructed by counting streamlines connecting regions defined by the DKT atlas[38,39], and compared to ground truth connectomes using Pearson correlation. The connectomes were normalized with respect to the number of streamlines per bundle. Evaluation was also performed using graph-theoretic measures implemented in the Brain Connectivity Toolbox[52]:
   - Density: Ratio of actual to possible connections.
   - Characteristic path length (CPL): Mean shortest path between nodes.
   - Global efficiency: Inverse of CPL, averaged over the network
   - Modularity: Degree of division into subnetworks.

### Tractography comparison

We evaluated five tractography methods and measured statistical significance between ATM-generated bundles and each of the other methods across the six evaluation metrics (similarity, coverage, validity, orientation, geometry, and connectivity) using a two-sided paired *t*-test.

1. *ATM*: ATM is applied to individual T1w images to generate 3000 streamlines per bundle.
2. *ATM-Population*: ATM is applied to a population-averaged T1-weighted image to generate 3000 streamlines for each bundle, which are subsequently transformed into each subject's native space.
3. *MRtrix with BundleSeg:* Whole brain probabilistic tractography using dMRI data followed by bundle segmentation using BundleSeg[48,53] with the Sherbrooke Connectivity Imaging Lab (SCIL) white matter (WM) atlas[54]. Specifically, the response function is estimated using the `dwi2response` command with the `dhollander` algorithm, followed by the estimation of fiber orientation distributions using `dwi2fod` with the `csd` algorithm. Probabilistic tractography was performed with `tckgen` using the iFOD1 (4th-order Runge–Kutta integration) algorithm, seeding from the white matter-gray matter interface provided in the TractoInferno dataset until 3 million streamlines are obtained.
4. *Atlas warping*: The SCIL WM tract atlas[54] is warped to each subject's space using a nonlinear transformation. The TractoInferno dataset, on which ATM was trained, used the SCIL atlas for bundle extraction from whole-brain tractograms; therefore, the SCIL atlas serves as a natural and appropriate reference in evaluating the performance of ATM.
5. *TractSeg*: TractSeg[28], a supervised deep learning method, is used to segment white matter bundles based on orientation maps derived from dMRI. For each segmented bundle, 2000 streamlines are generated. TractSeg was trained on a different tract atlas (XTRACT), so direct quantitative comparison is not possible due to potential differences in bundle definitions. For completeness, we provide a qualitative comparison in Supplementary Materials.

### Reporting summary

Further information on research design is available in the Nature Portfolio Reporting Summary linked to this article.

## Data availability

The TractoInferno data used in this study is available at https://openneuro.org/datasets/ds003900/versions/1.1.1. The SCIL WM atlas used is available at https://zenodo.org/records/10103446. Source data are provided with this paper.

## Code availability

The source code and trained models for this study are provided at Zenodo: https://zenodo.org/records/15792527. Software tools used in this work include Matlab R2023b; Python 3.10.0 with libraries including NumPy 1.23.5, PyTorch 2.5.1, Scikit-learn 1.2.1, DiPY 1.8.0, Matplotlib 3.6.3, Nibabel 5.2.0, and Scipy 1.9.3; FreeSurfer v7.3.2; MRTrix3 3.0.4; Surflce 13.4; FSL 7.0.6.4; DSI Studio (2023.12.06 "Chen" release); Workbench Command 1.5.0; and SciLPy 2.1.0, ANTs 2.5.4.

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

## Acknowledgements

This work was supported in part by the United States National Institutes of Health (NIH) under grants R01 MH125479, R01 EB008374, R01 MH133836, and R01 EB035160 (P.-T.Y.).

## Author contributions

Y.-F.T.: Methodology, investigation, visualization, data curation, writing—original draft, writing—review and editing. K.M.H.: Methodology, investigation, visualization, data curation, writing—review and editing. S.L.: Methodology. R.C.-W.P.: Resources. C.-M.T.: Funding acquisition, supervision, writing—review and editing. P.-T.Y.: Conceptualization, supervision, funding acquisition, resources, investigation, validation, writing—review and editing.

## Competing interests

The authors declare no competing interests.
