## [Transparent Peer Review file · Nature Communications]

Anatomy-to-Tract Mapping Infers White Matter Pathways Without Diffusion Streamline Propagation

Corresponding Author: Professor Pew-Thian Yap

Version 0:

Reviewer comments:

Reviewer #1

(Remarks to the Author)

This work presents a novel tractography method that does not require diffusion MRI data. Instead, a mapping from an embedded T1 volume to streamlines is directly performed. The contributions of this paper are two-fold: tractography without diffusion MRI, and “one-shot” streamline generation instead of the usual point-per-point process.

The strengths of this paper lie in the contributions, their motivations and the discussion of them. As mentioned by the authors, diffusion MRI is time-consuming to acquire which limits its adoption in the clinical setting. Moreover, tractography, due to its typical local nature, is plagued with false positives caused by challenging fiber configurations. The work presented by the authors addresses both issues by introducing a VAE trained on a small cohort of healthy young adults which maps directly from T1 to streamlines. The description of the ATM architecture, including the segmentation and streamline VAE modules, along with the training and inference processes, is detailed and comprehensive.

The method is well discussed and its limitations are adequately put forth. The authors mention, for example, that the method “is not immune to errors” which may “undermine the model’s reliability in clinical and research applications”. The authors take note that the method was only trained on HCP subjects which may not be representative of clinical data. The authors finally propose ways of improving the method such as including other modalities.

The paper however also suffers from severe limitations. (1) As T1 does not include directional information at the subject level, comparison with an atlas should have been provided. (2) The current validation performed on the reconstructed tractogram lacks comparison with competing methods. (3) The results are reported in a somewhat obtuse way. (4) Qualitative results could be improved. (5) The paper lacks the scope and rigour needed for a journal of this calibre.

(1) The method produces streamlines, which are inherently dependent on the orientation of white-matter structures, without access to such information. While the presented results are certainly impressive, this work either implicitly argues that T1 imaging contains orientational information or that tractography is not dependent on the orientation of a subject’s WM pathways. Indeed, machine learning is at its core pattern matching. Typical machine learning-driven algorithms match patterns of dMRI with angular directions. By mapping directly from T1 to streamlines, the algorithm has to match patterns from T1, which contains shape and volume (but not angular information) about the subject, to directions. The method may therefore instead infer local orientations from the population it was trained on, since this information is not available at the subject level. This raises the question: how different are the tractograms produced from registering a population-average atlas to the subject’s space? Is the VAE doing more than non-linear registration? Summarizing this long preamble, the evaluation of the method should have included a comparison with tractogram registration.

(2-3) Comparison with baseline tractography methods are only reported in text, in a somewhat incoherent manner as well. This obscures comparison with previous tractography methods and raises doubts on the performance of the proposed algorithm. Figures 3, 4, 5a-b, 6b and others in supplemental should provide a clear visual comparison with either or both conventional tractography algorithms and ML-driven methods. Quantitative results should include the standard deviation as well as the mean. Related to (1), the claim of the abstract that the results are subject-specific cannot be verified otherwise. Some metrics, such as bundle length, span, volume, etc. are often reported but never compared with either reference tracts

or competing methods, rendering them somewhat meaningless.

Tractometer (TM) results are reported and compared with existing methods, yet it is unclear how the Tractometer was used. TM typically works with two phantoms (ISMRM 2015 and FiberCup), was the method adapted to work on the HCP data as well? Or was the method used on the ISMRM2015 or FiberCup data? Was ATM trained on these phantoms as well or was generalization implied from the HCP data to either of these phantoms? Moreover, these phantoms do not provide the same bundles as the training data, how was performance evaluated? Finally, these results should be reported clearly in a table alongside competing tractography methods.

(4) The presented qualitative results are not displayed well enough for the reader to get a proper sense of their relevance. Figures 3a, S2, S3, S4 offer tractograms in three views but the resolution of the figures does not allow for meaningful comparisons. The choice of streamline coloring/shading makes it impossible for the reader to see small orientation changes along the streamline. This could make the bundle look more uniform/smooth than it is; considering the proposed method is claiming to reconstruct streamline this detail is important (see Dipy endpoints coloring vs Dipy local coloring for example). The color-scheme used for Figure 3b is also problematic and makes it hard to analyse the results presented. Figure 4 is more readable, but related to points 2-3, a comparison with other tractography algorithms would have been desirable. Only the coronal view is displayed in figure S5, which is suboptimal for association bundles like the arcuate fasciculus, anterior thalamic radiation or cingulum.

(5) Finally and most importantly, the paper lacks the scope required for publication in such journals as Nature Communications and would be better suited for a conference. One of the key motivations of the paper is to allow tractography on clinical data where diffusion MRI is not acquired, yet results are mostly presented on 8 research-quality healthy young-adult subjects. Related to (1), the actual usefulness of reconstructed streamlines on downstream, research or clinically relevant tasks is never addressed, especially compared with atlas registration. Results should be reported on cohorts and the potential for societal impact should be unequivocally demonstrated.

Here are some additional major comments:

The bundle adjacency score is a distance in millimeters similar to a Hausdorff distance. I don't understand how this is converted to a percentage. The BA metric should be kept as millimeters.

When reporting Overlap/Overreach, the F1-score should always be mentioned first, as OL/OR are more specific.

High geometric agreement with ground truth (Fig 4): I believe it would be crucial to assess the local orientation of the streamline. I recommend computing Tract-Oriented Density Map (TODI) and computing an Angular Correlation Coefficient (ACC) to demonstrate that the global geometry (overall voxel map) and the local geometry (streamline orientation) are in agreement.

Figure 5 should be removed or severely corrected. It is very hard to interpret similarity metrics at the scale of all associations or the whole brain. This figure takes up a lot of space. Instead, I would recommend the authors group association, commissure, and projection in their Figure 6 to show these groupings.

If I understand correctly, all these graphs have uncertainty (since computed across subjects, or inter-session), so box plots would be more appropriate or histograms with error bars. Not showing the STD is problematic as it is crucial to understand the proposed method's performance (and stability).

When looking at the results (especially Figure 6), the F1 score seems incredibly low (almost never above 50%). Showing so many bundles with poor scores is not reassuring. Since the comparison methods are mentioned only in the text, it is hard to put into context if the proposed method is an improvement or merely an equivalent method to existing methods.

I am not sure why the authors are showing graph theory measures for bundles in Fig 6. This is not only hard to interpret, but there are no baselines from any other method. It is very hard to know at a glance using the figure if the proposed method is inferior or superior to anything else available or if the value even makes sense.

Here are some minor corrections:

Line 158: Numerical results based on based on

Fig 5: Streamline validity ratio should be capped at 100

Update citation for Leon Cai: https://direct.mit.edu/imag/article/doi/10.1162/imag_a_00259/123726/Tractography-from-T1-weighted-MRI-Empirically

(Remarks on code availability)

The code provided for the publication deviates from expected standards for open-source code in the context of a high-impact journal.

While the code is likely functional (locally on the author's computer), it lacks essential elements for easy adoption and reproducibility. The absence of a requirements file and setup.py makes it tedious to install dependencies, and the lack of

documentation and tests hinders understanding and verification of the code's functionality. Additionally, the code's formatting and structure could be improved to adhere to established best practices (pep8 at least).

The lack of an argparser or --help parameters make it challenging to run and the expectation of undisclosed file structures for loading and saving makes it impossible to actually run the code without trials/errors.

Considering the importance of open-source code in today's research landscape, especially for a publication in Nature Communications, it is crucial to provide clear documentation, demonstration data, and well-formatted scripts with user-friendly CLIs.

The provided code is insufficient to reach the standard of open-source code, there is too many barriers to reproduce or even simply try the proposed methods.

Reviewer #2

(Remarks to the Author)

(Remarks on code availability)

See the complete review.

Reviewer #3

(Remarks to the Author)

Manuscript ID: NCOMMS-24-83268-T

Title: Anatomy-to-Tract Mapping: Inferring White Matter Pathways Without Diffusion Streamline Propagation

First Author: Tan, Yee-Fan

This manuscript presents "Anatomy-to Tract Mapping (ATM)", a deep learning framework that generates bundle-specific streamlines directly from commonly acquired anatomical MRI, bypassing diffusion-MRI based tractography. The authors convincingly show that ATM provides anatomically realistic reconstructions, and claim that this method overcomes limitations of traditional tractography (crossing fibers, partial volume effects, etc.). The data is trained on, and framework validated on, Human Connectome Project datasets.

However, I have two major criticisms or points of weakness. First, is innovation and advances compared to existing literature studying pathways using T1w images only – where it is not clear what sets this apart, or how this network also is not effected by traditional tractography challenges (when it is indeed trained on traditional tractography methods). Second is suprisingly poorer results when looking at individual bundles (instead of combining all association/projection/commissural pathways).

I will start with strengths - as there are several strengths and novelties:

1. Innovation: paradigm shift in tractography by relying on commonly acquired, geometrically accurate, and higher resolution T1-weighted images for direct bundle segmentation.
2. Scientific impact of application when diffusion data is unavailable, or application on a population-based reference with potential pathway-specific analysis on non-diffusion-MRI studies (PET, fMRI, etc.)
3. High reproducibility, and strong assessment and comparison with standard bundle segmentation benchmarks.

Major Points:

First, is the innovation and separation from existing literature, and methods that infer white matter pathways without diffusion data. Indeed, the authors do cite other literature that generates streamlines from anatomical image alone (Cai et al., *Imaging Neuro*, 2024) – yet statements such as "for the first time, generates complete, bundle specific streamlines directly from anatomical MRI, entirely bypassing the limitations of streamline propagation." suggest that this is the first work to do so. It is quite possible that the separation is in the modifier "bypassing limitations of streamline propagation", yet again, the data in the current manuscript was trained with (and validated with) streamlines that do in fact use streamline propagation and suffer from these limitations. There are also other segmentation methods that generate WM labels without streamline propagation (or even streamlines) – some quick examples include (<https://onlinelibrary.wiley.com/doi/full/10.1002/hbm.70063>; <https://aapm.onlinelibrary.wiley.com/doi/10.1002/mp.15495>) which generate JHU labels or TractSeg labels with T1 only. So, the major advance/innovation is in the network itself, and how these streamlines are sampled/generated.

Second, this would be extremely convincing if the results were compared to simply warping a population atlas of streamlines to an individuals T1. I believe that simply doing this, rather than an entire deep learning network, would lead to much higher values of bundle similarity, bundle overlap, and minimal overreach. I'm quite surprised by individual bundle overlap of ~30-50% (Figure 6, SF5), bundle similarities of large bundles <60% (AF_left, MLF, CC6) or low (<60%) or even no correlation. Most qualitative and quantitative analysis seems to be on "commissural", "association", "projection" fibers together (which give high overlap, etc., but occupy a majority of the brain white matter) instead of individual bundles (which seem to have much poorer measures) – given in Figure 6. I would be very surprised if warping a bundle from an atlas (AF_left, for example) gave a lower overlap than the ~35% from this network. Similarly, Figure 6 (the individual bundle

analysis) is titled “smaller bilateral and commissural bundles” but these are indeed quite large bundles, where many bundles occupy a large fraction of the white matter, and thus I would expect generally high overlap/Dice/etc.

Minor points:

1. This would be additionally convincing, and a stronger message for NCOMMS if generalizability to non-HCP data were shown. As this relies on T1 only, most clinical scanners can acquire quite acceptable T1w images that should be comparable to HCP T1w images, whereas most scanners cannot acquire the high resolution, high b-value data if diffusion MRI is needed.
2. I would like to see qualitative results (pathways) of individual bundles within the main manuscript (similar to F3 and F4, but for individual bundles) rather than supplementary material...as combining the entire PROJ/COMM/ASSOC pathways is such a large region that we would expect even simple networks to have high overlap. Visually, results of SF5 appear that they should have much higher overlap than only 30-50% indicated in table S5 (AF_LEFT, for example with overlap 35%)
3. I was hoping for more discussion on what it means that we can do tractography on T1w images? What does it mean about the relationship between the info in T1 (cortical folds, scale of 100's of mm) versus the microstructure we get from diffusion (scales of micrometers) – can we get all the tractography information from the large scale structure? If so, where does that place the importance of acquiring diffusion data? I think a high level discussion would be of interest to NCOMM readers.
4. Similar to major point #2, I don't believe most of the qualify as “smaller” bilateral or commissural bundles. These are indeed the largest functionally relevant bundles that are identified in the brain, spanning multiple lobes and gyri.

(Remarks on code availability)

Reviewer #4

(Remarks to the Author)

Key results:

In this manuscript, the authors propose a neural network-based approach to streamline tractography, called Anatomy-to-tract mapping (ATM), that relies solely on T1w images as input data. Such an approach comes with the theoretical potential of analysing white matter bundles without relying on more demanding diffusion MRI (dMRI) acquisitions, which would allow for tractography to be applied in a wider range of clinical applications in the absence of dMRI data. Beyond this, the investigation of streamline generation implemented as a global process is interesting as it can by design avoid typical challenges faced by the step-wise streamline propagation used in local tractography approaches. In this context, combining the generative streamline creation process using variational autoencoders (VAEs) with the processed information of gridded image data in an end-to-end trainable fashion is a novel approach that has not been investigated so far to my knowledge.

Validity:

While the proposed method, ATM, is interesting and bears the potential to be useful in different applications, the goals of the paper could be pointed out more concisely. To my understanding, the authors identify two main limitations of conventional tractography methods that they aim to address with ATM:

1. Challenges introduced by local, step-wise streamline propagation techniques,
2. Limitations arising from the dependency on dMRI as input data (e.g., limited quality, limited availability).

Yet, it is not clear whether the manuscript is intended to show evidence for ATM specifically improving over local tractography techniques or for ATM outperforming dMRI tractography methods or if, alternatively, this should be a proof of concept that ATM-like tractography can produce streamlines of a certain quality. Specifically, the authors do not explicitly define any performance goal for ATM that should be reached for them to arrive at a particular conclusion.

Depending on the concrete definitions of goals for the paper, the experimental methodology could be improved. From my perspective, addressing both of the above limitations of common tractography methods is not easy since both are likely to affect performance in opposite directions. While addressing local tracking limitations could be expected to improve results compared to established local-tracking approaches, not relying on dMRI as input should result in a drop in tractography quality (e.g., Cai et al. (2024)). If the authors intend to address both aspects, the experiments should optimally be designed to show separate evidence for both claims, i.e.,

1. Showing evidence for improved tractography by avoiding local tracking,
2. Showing evidence for sufficiently good tractography even without relying on dMRI data.

In my opinion, the experiments are currently focused on showing that ATM's streamline bundles are sufficiently close to dMRI-based tractography (although also the evidence for this claim could be expanded), but there are no experiments presented that directly assess a potential superiority of the non-local streamline creation over local approaches.

Finally, the presented experiments assess various metrics of streamline and bundle quality with the intention to provide evidence for a “competitive performance” of ATM (line 199), yet, the process to reach this conclusion based on the presented experiment results could be explained in more detail. In my opinion, it would be possible to either explicitly define performance goals based on ranges of metric values that are generally considered to be “good” (would need a rationale or references), or to compare ATM to reference methods in a reasonably close experimental setting. In most of the conducted experiments, the authors reference achieved metric values by other approaches from literature, however, it needs to be discussed in the manuscript why the authors think that the experimental settings allow for a fair and conclusive comparison. In particular, all experiments for ATM are conducted on the TractSeg dataset (of 105 subjects) and another Human Connectome Project (HCP) subset for test-retest data, while, for example, some of the compared methods (GESTA, Track-to-Learn, DeepTract) have been assessed on the ISMRM 2015 Tractography Challenge synthetic data phantom. That means

that both, input data (in-vivo vs. synthetic) and labels (different bundle definitions and different sets of bundles) differ between ATM and the compared methods which is likely to affect the Tractometer scores. The same can be said about other assessed measures for “similarity”, “geometry”, “connectivity”, and “test-retest consistency”. A direct comparison between methods on the same dataset would be preferable to rule out any effects of such experimental differences on the comparison scores (e.g., applying ATM to the ISMRM phantom dataset, or applying other methods to the datasets in this manuscript, or applying ATM and other methods to additional external datasets). In my opinion, it is not possible to unambiguously conclude that ATMs performance is “competitive” in absence of a concrete reference from another method. In fact, it is not clear what value range the authors define as “competitive”.

Originality and significance:

In general, the question of the potential of tractography based on T1w images is of interest for the field. Cai et al. (2024) investigated this question by analysing the “clinical viability” of their T1w-based streamline propagation approach and find that some effects found through dMRI tractography can be replicated while others cannot. Thus, there is room for further evidence for or against the potential of such alternative tractography methods. With this in mind, the comparison of ATM with other relevant methods, CoRNN (Cai et al. (2024) as a T1w-based approach but also relevant classical dMRI-based methods, could be a valuable contribution to the field. Particularly, the question of differences in information content included in the two modalities and their importance for tractography could be discussed in the context of this manuscript, too.

However, in the present state, the latter aspect is not included in the discussion and concrete comparisons between ATM and T1w- or dMRI-based methods are lacking in this work.

On a further note, also the technological aspect of circumventing streamline propagation is novel and is appealing to, theoretically, overcome common challenges of local tractography, as the authors point out. Thus, the architecture of ATM could not only be used for T1w-based tractography, but could potentially also prove useful as a dMRI-based version with hypothetical improvements over a step-wise tractography algorithm. While the proposed architecture of ATM could be evaluated and discussed in these regards with important insights for the field, in the current version of the manuscript, it is neither directly assessed in experiments nor discussed based on results from dedicated experiments.

Data and methodology:

Methodology:

1. As mentioned, in the current form of the manuscript, the absence of reference methods evaluated in the same experimental settings make it difficult to draw strong conclusions on the performance of ATM based on the reported experiment results.

Depending on the goals of the paper, one could consider adding comparisons with reference methods like a. other ML-based tractography methods, b. other bundle-specific methods, c. other T1w tractography methods, d. other conventional dMRI-based methods.

2. Since ATM is a neural network approach, testing the method with an external test dataset would be helpful to assess the generalizability of the approach. The dataset TractoInferno (Poulin et al. (2022)) is specifically designed for such experiments, comes with curated bundles, offers multi-site data as well as benchmarks from reference methods and should be taken into account to evaluate ATM.

Beyond this, there are several aspects of the methods that remain unclear and could be expanded upon or added:

1. Different experiment series are followed. Which dataset was used for which experimental analysis is sometimes not easy to understand from reading the text. There are different sets of bundles (a. 72/72 bundles, b. 48/72 bundles), different sets of subjects (105 subjects, b. 45 subjects), and different numbers of generated streamlines (N=1/8/12/16/20k). I would suggest a graphical overview to make the experiment structure clearer, or to give distinct names to those experiment lines that can easily be referenced in the text. For example, it is not clear how many streamlines were used for the test-retest evaluations in Tables S7 and S8, and what data (subjects and bundles) the model was trained and evaluated on.

2. The terminology when referring to “bundles” is sometimes confusing. I would suggest to use the word “bundle” only to refer to an anatomically defined bundle (and tractograms resembling a single anatomical bundle) and otherwise use the terms “merged bundles” or “combined bundles” or “whole brain tractogram” to refer to combinations of bundles. In the same context, it is not clear what the authors mean by “smaller bundles” (e.g., lines 73, 91, 104). Are those smaller in volume or are those “individual bundles” (regardless of their size)?

3. What is the rationale to group the individual bundles in three categories and report results per category (lines 71-72)? What is the rationale for reporting bundle-wise measures only for a subset of 48/72 bundles? How were these 48 bundles (lines 91-92) chosen?

4. The process of computing metrics for the three bundle categories should be detailed. It is unclear whether the metrics are computed per bundle and then averaged or whether the bundles are first merged and subsequently, metrics are computed. For the Tractometer measures overlap (OL) and overreach (OR), this can make a difference. How can it be explained that the average OL of the individual bundles in Table S5 (48/72) is lower than the OL for the three categories in Table S3?

5. How was the number of sampled streamlines chosen in training and in testing / evaluation? Are 8k streamlines enough to represent all bundles in a category? Assuming approximately 20 bundles per category, there would be about 400 streamlines per bundle. For some, this might be enough to capture all anatomical detail, but for others it might be insufficient. When sampling per category, is it possible that some bundles are underrepresented in the sampling process in training? In general, could it make sense to conduct an experiment in the beginning to determine the optimal number of streamlines to sample in training and in testing and then rely on those numbers in the subsequent experiments? Why are a different number of streamlines (N=1k) chosen for the experiments with 48/72 bundles (l. 92)? Are the N=1k streamlines sampled for training or for testing?

6. Different aspects in the description of the streamline generation need to be reported in more detail to allow for reproducibility of the results:

a. What value was chosen for the threshold length “ t_{l1} ” (l. 329)? Was this value bundle-specific or global? How was “ t_{l1} ” chosen?

- b. Is there an individual KDE for each bundle? Is KDE fitting a separate process after completed training of the neural networks? What data is used for fitting?
7. How were ML-related hyperparameters (e.g. learning rate, number of epochs, loss weights) chosen? Were their values explored? After training is complete, which model state is chosen for evaluation?
 8. In Fig. 2a, the streamline encoder has the bundle label as input, while in Fig. 2b, the encoder lacks that input.
 9. Lines 294-295: In the formula " $z_s = \mu_s + \epsilon \times \sigma_s$ ", is " ϵ " a scalar or a vector (it seems to be a vector since it is printed in bold face)? Could the operation denoted by " \times " be specified? If " ϵ " is a vector, I assume " \times " should be an element-wise product and " ϵ " follow a multivariate normal distribution?
 10. Line 286: Is the bundle label an input of D_s ? In Fig. 2a, it is not.
 11. Line 300: Do the ground truth bundle masks cover white matter only or both, white and gray matter (i.e. with streamline endpoints)?
 12. Line 82: Are the 45 retest subjects part of the 105 subjects? If so, was there data leakage (since there are only 8 test subjects)?
 13. Lines 107-119 and Fig. S6: What are the tSNE parameters used for the visualizations? Do the subfigures show the results of different ATM models? If so, how was each of them trained and tested? Why are not all bundles shown in one tSNE plot? This might be relevant since the inputs to the tSNE algorithm affect the obtained distribution of data points in the plot.
 14. According to tractometer.org and Renaud et al. (2023), the definition of a valid streamline is that of a streamline that belongs to a valid bundle. They also point out that the definition of a valid bundle depends on the procedure to segment a bundle which is a choice by the operator. Could the authors detail the criteria for a valid streamline in the methods section? (Are there endpoint masks? Is there a whole-bundle envelope mask? Is it based on bundle recognition (e.g. RecoBundles)?) What was the tractometer version the authors used? Is the tractometer version / methodology used to score ATM the same as those used in the compared literature?
 15. Several of the reported measures to score ATM are compared with methods from literature. Some relevant information for those external experiments should be reported and their suitability be commented on if possible:
 - a. How were the respective benchmark methods selected and why are they suitable? For example, CtTrack, Track-to-learn, and DeepTract are whole-brain approaches. As such, they might be more prone to creating invalid streamlines than a bundle-specific approach like ATM. Why not compare valid/invalid streamlines ratios to a bundle-specific approach (e.g., Poulin et al. (2018), or Wasserthal et al. (2019))?
 - b. What are the bundles that the results of the compared methods are reported on? Are they the same as those used for ATM? For example, CoRNN reports geometric measures like volume, length, span and surface area for 39 bundles identified by RecoBundlesX, while ATM reports measures averaged over bundle categories consisting of 72 bundles or over 48 individual bundles. Is this difference expected to impact results and conclusions?

Conclusions:

As mentioned before, the experiment methodology is limited by the lack of reference methods. This makes it challenging to conclude how the performance of ATM compares to other relevant methods. Beyond that, some conclusions seem to be general and could be investigated more in-depth, connected to concrete results, or discussed in more detail:

1. Lines 97-98: Did the authors explicitly test whether the fornix is generated completely? Table S2 does not list the fornix as part of the individually assessed bundles. Why was the fornix not included to support that statement with numerical evidence?
2. Lines 114-117: I am not convinced that the fact alone that the latent representations of streamlines show up as clusters in the tSNE plots allows for the direct conclusion that local streamline features are captured. Since the bundle label is an input to the streamline encoder, such clustering could theoretically also stem from that label alone, even though streamline decoding would not work as well in that case. Since the training is performed with labels of the individual bundles, even the visualization for the three bundle categories could look like in Fig. S6. Isn't the clustering rather a necessary than a sufficient criteria for capturing local streamline features?
3. Lines 126-129: It is said that "ATM consistently produces bundles with competitive" scores, but it seems that only results for $N=8k$ in table S3 are reported in the text. Noting that at least the overreach scores are a lot higher for other choices of N , it is unclear what is meant by "consistently". It would also be helpful if the range for "competitive" scores could be defined explicitly. For example, the overlap range for GESTA (line 133) is reported as mostly higher than that of ATM (line 127). Finally, in my opinion, the added value of Fig. 4 in assessing the overreach score (lines 127-128) based on single slice projections of single subject images is limited. It could be more helpful to show a plot of the numerical overlap and overreach scores of ATM and of the compared literature instead.
4. Lines 170-173: In general, increasing overreach does not indicate improved spatial matching as claimed in line 172. But in fact, overreach seems to decrease for $N>12k$, which is unexpected to me. Typically, I would expect the overreach to increase with a larger number of streamlines created. Can this effect be commented on?
5. Lines 174-176: Can it be explained what the range is for "relatively stable" (line 175) geometric fidelity? The values for whole-brain volume and surface area seem to vary a lot and particularly for $N=16k$, the variations for both measures are large, too. Can the authors comment on this observation?
6. Line 178: What is the interpretation of the overlap and overreach scores of the test-retest experiment? In view of the overlap values in Table S3 being mostly larger, what do these values mean for ATMs capability to capture subject-specific characteristics reliably? Maybe a benchmark method could help to assess if these scores are good or bad.
7. Line 227: In the given experimental paradigm without a reference method, is it clear whether the subject-specific anatomy is captured well enough? What would the performance measures for a subject look like when streamlines are created from a "template" T1w image? Can this be tested somehow?

Suggested improvements:

Most importantly, the experiments should incorporate at least one reference method for direct comparison on the same

evaluation dataset. Additionally, experiments on an external dataset should be considered to test the generalization performance of ATM. The dataset TractoInferno (Poulin et al. (2022)) should be considered to be used (see also Legarreta et al. (2023) for an example) as it provides data from multiple sites with curated bundles and benchmarked reference methods.

References:

The following references should be added:

1. Cai et al. (2024)
2. Poulin et al. (2022)
3. Poulin et al. (2018)

The following references should be updated:

1. Ref 11: Cai et al. Convolutional-recurrent neural networks approximate diffusion tractography from T1-weighted MRI and associated anatomical context. *Medical Imaging with Deep Learning*, PMLR 227:1124-1143 (2024).

Clarity and context:

The introduction covers the dMRI tractography problem, lists ML tractography methods briefly, highlights CoRNN and then describes ATM over two paragraphs. In my view, the background should be expanded with other relevant methods that share parts of their concept with ATM. ATM builds upon a. Bundle segmentation, b. Variational autoencoders (VAEs) for streamlines, and specifically streamline generation, c. Streamline post-processing and validation, d. Bundle-specific tractography. Adding background on VAEs for streamline generation (including GESTA, maybe also FINTA) as well as the introduction of bundle-specific tractography as a method to address some of the mentioned challenges of local streamline propagation (e.g., including Wasserthal et al. (2019)) would provide more context for ATM.

Overall, the idea of T1w tractography could be introduced and discussed in a more balanced way. In this context, the arguments around the value and potential of T1w-based tractography introduced in Cai et al. (2024) could be picked up and put in relation with findings in this manuscript. In its current form, the manuscript seems to talk overly positive about tractography based on T1w data without discussing any limitations found during the experimental evaluation of ATM or concerns about potential limits of tractography based on T1w data in general. In the end, it remains elusive how the performance of ATM compares to 1. dMRI tractography, 2. CoRNN (as a T1w tractography method) or other ML-based streamline propagation approaches, 3. GESTA as a bundle-specific, generative, non-dMRI tractography method. It would be helpful to define explicit goals to point out which of these hypotheses, if any, should be tested in the experiments of this manuscript.

The abstract could be augmented with concrete descriptions of the experimental design and most important results. In the present state, the focus seems to be on theoretical advantages of the proposed architecture of ATM, but lacking descriptions of the methodology of the paper.

Minor comments:

1. The first two paragraphs of the results section might better fit in the methods section.
2. Line 304: typo "... that constrain(t)s ..."
3. Line 289: define "s" as $(x, y, z)_i$; define "S" and "X"?
4. In line 310, variable "N" is referenced, but doesn't appear in formula (3). Also, does "N" refer to the number of streamlines per batch (i.e. "batch size")?
5. Formula (3) (Cross-entropy): Should the summation be over classes (i.e., "bundles") instead of samples (i.e., "streamlines")? Should c_i be an indicator function for the class c_i ?
6. Line 316: should formula for $B(i)$ run to "K-1"?
7. Unify loss formulae? (some are per batch, some are per streamline)
8. Unify use of acronym "T1w". Sometimes it is written as "T1-weighted".
9. Line 123: add reference to Fig. 5?
10. Line 158: typo "Numet(r)ical results ..."
11. Line 175: avoid use of "significant" in absence of statistical tests?

List of references:

1. LY Cai, HH Lee, GW Johnson, ..., BA Landman. Tractography from T1-weighted MRI: Empirically exploring the clinical viability of streamline propagation without diffusion MRI. *Imaging Neuroscience* (2024) 2: 1-20.
2. E Renaud, A Theberge, L Petit, JC Houde, M Descoteaux. Validate your white matter tractography algorithms with a reappraised ISMRM 2015 Tractography Challenge scoring system. *Scientific Reports* 13, 2347 (2023).
3. P Poulin, G Theaud, F Rheault, E St-Onge, ..., M Descoteaux. TractoInferno - A large-scale, open-source, multi-site database for machine learning dMRI tractography. *Scientific Data* 9, 725 (2022)
4. P Poulin, F Rheault, E St-Onge, PM Jodoin, M Descoteaux. Bundle-Wise Deep Tracker: Learning to track bundle-specific streamline paths. In *Proceedings of the International Society for Magnetic Resonance in Medicine (ISMRM-ESMRMB, 2018)*.
5. J Wasserthal, PF Neher, D Hirjak, KH Maier-Hein. Combined tract segmentation and orientation mapping for bundle-specific tractography. *Medical Image Analysis* 58, 101559 (2019).
6. JH Legarreta, L Petit, PM Jodoin, M Descoteaux. Generative Sampling in Bundle Tractography using Autoencoders (GESTA). *Medical Image Analysis* 85, 102761 (2023).

(Remarks on code availability)

Version 1:

Reviewer comments:

Reviewer #1

(Remarks to the Author)

The authors thoroughly addressed all reviewers comments and improved the quality of the manuscript.

Small comments about RecobundlesX in the manuscript:

1) Depending about the Scilpy version (≥ 1.6), you maybe have used BundleSeg instead of RecobundlesX (we made that clearer in newer release), can you verify the --help of the script you used and check if you actually use RecobundlesX or if the script refers to BundleSeg?

(and update citation if it is in fact BundleSeg: <https://arxiv.org/abs/2308.10958>)

2) RecobundlesX or BundleSeg are segmentation methods, not a WM atlas (in fact I use it with various WM atlas). Could refer to the Zenodo where you found the atlas to be more precise.

From the look of it, it is either <https://zenodo.org/records/7950602> or <https://zenodo.org/records/10103446> (it is 99% identical, but one contains the config file for RecobundlesX and the other config files for BundleSeg). You could say these are SCIL (Sherbrooke Connectivity Imaging Lab) WM atlas?

- Francois Rheault

(Remarks on code availability)

Reviewer #2

(Remarks to the Author)

(Remarks on code availability)

Reviewer #3

(Remarks to the Author)

Please see attached document with my review. The submission system does not allow me to provide screenshots (showing TractSeg successfully runs on TractoInferno Subject ID: 1135) - so all comments are attached in PDF form.

I have suggested revision. In short, my concerns relate to (1) clarifying what is being compared (2) ATP population benchmark (3) fair benchmarking (and proper running of benchmarking algorithms) (4) advantages over previous methods

(Remarks on code availability)

Version 2:

Reviewer comments:

Reviewer #3

(Remarks to the Author)

I applaud the effort to address all concerns - the authors have well-addressed all comments, and I recommend accepting as-is in Nature Comm.

(Remarks on code availability)
